*Cambridge Prisms: Global
Mental Health*

## Research Article

Onie S, Bandara P, Kumar GA, Spittal M, Page A,
Vijayakumar L, Pirkis J and Dandona R (2025).
Trends in suicide among adolescents aged
14–17 years in India: 2014–2019. *Cambridge
Prisms: Global Mental Health*, **12**, e90, 1–7

Suicide; India; adolescents; suicide prevention;
National Crime Records Bureau (NCRB);
Adolescent Health Strategy (IAHS); National
Suicide Prevention Strategy (NSPS)

**Corresponding author:**
Vikas Arya;
Email: arya.v@unimelb.edu.au

# Trends in suicide among adolescents aged 14–17 years in India: 2014–2019

Vikas Arya[1] , Gregory Armstrong[1], Caley Tapp[2,3], Sandersan Onie[4,5],
Piumee Bandara[6,7], G. Anil Kumar[8], Matthew Spittal[9], Andrew Page[7],
Lakshmi Vijayakumar[10], Jane Pirkis[1] and Rakhi Dandona[1,8,11]

[1]Centre for Mental Health and Community Wellbeing, Melbourne School of Population and Global Health, The University
of Melbourne, Melbourne, VIC, Australia; [2]School of Public Health, University of Queensland, Brisbane, QLD, Australia;
[3]Queensland Centre for Mental Health Research, Wacol, QLD, Australia; [4]Black Dog Institute, UNSW Sydney, Sydney,
NSW, Australia; [5]Department of Global Health and Social Medicine, Harvard Medical School, Boston, MA, USA;
[6]Population Health Sciences, Bristol Medical School, University of Bristol, Bristol, UK; [7]Translational Health Research
Institute, Western Sydney University, Sydney, NSW, Australia; [8]Public Health Foundation of India, Gurugram, India;
[9]Melbourne School of Population and Global Health, The University of Melbourne, Melbourne, VIC, Australia; [10]Sneha –
Suicide Prevention Centre, Voluntary Health Services, Chennai, India and [11]Institute for Health Metrics and Evaluation,
University of Washington, Seattle, WA, USA

## Abstract

This study investigates the epidemiology of adolescent suicide in India, addressing the limited
research on the subject. Data on adolescent suicide (14–17 years) by sex and state were obtained
from the National Crimes Records Bureau for 2014–2019, which included acquiring unpub-
lished data from 2016 to 2019. Crude suicide rates for the period 2014–2019 were calculated by
sex and state. Rate ratios (RRs) by sex and state were also calculated to assess changes over time,
comparing suicide rates from 2017–2019 to 2014–2016. Female adolescent suicide rates, which
ranged between 9.04 and 8.10 per 100,000 population, were consistently higher than male
adolescent suicide rates, which ranged between 8.47 and 6.24 per 100,000 population. Compared
to the first half of the study period (2014–2016), adolescent suicide rates significantly increased
between 2017 and 2019 among less developed states (RRs = 1.06, 95% uncertainty interval [UI] =
1.03–1.09) and among females in these states (RRs = 1.09, 95% UI = 1.05–1.14). Male suicide
rates aligned with global averages, while female rates were two to six times higher than in high-
income and Southeast Asian countries. Findings highlight the urgent need for comprehensive
surveillance and targeted suicide prevention strategies to address this critical public health issue.

## Impact statement

Adolescent suicide is a critical yet under-researched public health issue in India. This study
provides the first comprehensive analysis of suicide rates among adolescents aged 14–17 years in
India between 2014 and 2019, addressing a significant data gap by examining trends at both
national and state levels. The study found that adolescent suicide rates among females are higher
than those among males. Furthermore, compared to 2014–2016, adolescent suicide rates
increased between 2017 and 2019 among less developed states, particularly among females
within those states. This study has direct implications for national policies, including India's
Adolescent Health Strategy and National Suicide Prevention Strategy. It highlights the need for
enhanced surveillance of adolescent suicidal behavior, integration of gender-sensitive interven-
tions in schools and improved mental health services. Additionally, the study advocates for more
comprehensive suicide data reporting by the National Crime Records Bureau, recommending
disaggregation by state, sex and refined adolescent age groups (10–14 and 15–19 years).

## Introduction

Suicide, defined as intentionally ending one's own life (Turecki et al., 2019), is a major global
public health issue, with over 700,000 deaths reported worldwide in 2021 (WHO, 2025).
Approximately 73% of these suicides occur in low- and middle-income countries (LMICs).
The global suicide rate was estimated at 8.9 per 100,000 population in 2021, with higher rates
observed in the South-East Asia Region (10.1 per 100,000 population) (WHO, 2025).

Suicide is among the leading causes of death worldwide among adolescents, typically defined
as individuals aged 10–19 years (WHO, 2014; Glenn et al., 2020). India, which accounts for ~28%
of the world's suicides, including 25% of all male and 37% of all female suicides globally (India

State-Level Disease Burden Initiative Suicide Collaborators, 2018; Arya, 2024), has a notable gap in understanding suicide rates among adolescents. Studies in India often group individuals into age ranges of 0–14 and 15–29 years (Arya et al., 2018; Jena et al., 2024), which limits insights into adolescent-specific suicide rates, as neither of these age groups exclusively represents adolescents. This is especially important as adolescence represents a critical transitional period in the life course, and adolescent suicides can have devastating emotional and economic consequences for families and communities (Balaji et al., 2023).

Approximately one-fifth of India's population is accounted for by adolescents, presenting a significant opportunity to positively impact the country's social and economic landscape (Ministry of Health and Family Welfare, 2014a). The two policies best suited to address adolescent suicides in India are the National Suicide Prevention Strategy (NSPS) (Ministry of Health and Family Welfare, 2022) and the India Adolescent Health Strategy (IAHS) (Ministry of Health and Family Welfare, 2014a). The NSPS highlights that most suicides in India occur among youth and middle-aged adults, while the IAHS references suicide as a concern among individuals aged 15–29 years. However, neither policy document, despite drawing from national data sources, specifically addresses suicide rates or trends among adolescents in India, likely due to insufficient data on adolescent-specific suicide rates and trends.

Effectively tackling adolescent suicide requires a comprehensive understanding of the issue within this specific age group, rather than including them in a broader category. This study aims to bridge this crucial gap by examining trends in adolescent suicide rates in India from 2014 to 2019 at both national and state levels, disaggregated by sex.

## Methods

This study employed a descriptive-analytic ecological design using publicly available secondary data on suicides to examine the national- and state-level trends and differentials in adolescent suicide rates in India between 2014 and 2019, disaggregated by sex and state development status. The National Crimes Records Bureau (NCRB) is the national administrative data source for suicide in India based on the cases reported to police (NCRB, 2001–2022). It collects and publishes suicide data from each state and union territory of India. The NCRB data are stratified by different variables, including state and sex. Up until 2013, the NCRB published suicide data by five different age groups, namely 'Up to 14 years', '15–29 years', '30–44 years', '45–59 years' and '60 years and above'. In 2014, the 15–29 years category was further divided into '14–17 years' and '18–29 years' categories, presenting a unique opportunity to explore suicide rates among adolescents in India aged 14–17 years at the national and state levels. However, the NCRB discontinued publishing age-stratified data by state from 2016 onwards. We obtained unpublished NCRB age-stratified suicide data by state from 2016 to 2019 under the Right to Information Act (Roberts, 2010). Data from 2020 onwards were not available from the NCRB. Since the last Census of India was conducted in 2011 and updated official population estimates disaggregated by age and state were not available for the study period, we used state-wise annual population estimates for adolescents aged 14–17 years from the Global Burden of Disease (GBD) study for the years 2014–2019 (GBD, 2021). These GBD population estimates were used solely as denominators for calculating suicide rates. The GBD dataset was not used for any suicide count data or trend analysis.

We calculated crude suicide rates and 95% confidence intervals assuming a Poisson distribution for suicide counts for each year from 2014 to 2019, stratified by both sex and state. Following the methodology of a previous study on suicide in India, we also calculated crude suicide rates for two distinct groups of Indian states based on their socioeconomic development level: 'less developed' (including the states of Bihar, Chhattisgarh, Jharkhand, Madhya Pradesh, Odisha, Rajasthan, Uttar Pradesh, Arunachal Pradesh, Assam, Manipur, Meghalaya, Mizoram, Nagaland, Sikkim, Tripura and Uttarakhand) or 'more developed' (including the states of Andhra Pradesh, Telangana, Jammu and Kashmir, Himachal Pradesh, Punjab, Haryana, Gujarat, Maharashtra, Goa, Karnataka, Kerala, Tamil Nadu and West Bengal) (Dandona et al., 2017; Arya et al., 2018). For the purpose of this analysis, Jammu and Kashmir was considered an undivided state (Supplementary Material, pp. 3–8). The classification of states as 'less developed' and 'more developed' is used to assess whether patterns in adolescent suicide differ by the level of development, thereby helping to identify regional disparities and inform more targeted prevention strategies.

To determine whether there were significant changes in suicide rates, we calculated rate ratios (RRs) of crude suicide rates, stratified by sex and state, by dividing crude suicide rates of the last 3 years of the study period (i.e., 2017–2019) by the first 3 years of the study period (i.e., 2014–2016). We calculated RRs separately for 'less developed' and 'more developed' state categories. Since RRs are derived statistics based on estimates from multiple time points, all RRs are presented with 95% uncertainty intervals (UIs), which were obtained using Monte Carlo simulations with Ersatz software (Barendregt, 2009) (Supplementary Material, p. 9). The ErRelativeRisk function was employed, assuming a normal distribution for the natural logarithm of RR with a standard deviation of SE[ln (RR)]. This process involved 1,000 iterations to ensure the convergence of model outcomes. Analyses were conducted in MS Excel, STATA 15.1 and Ersatz 1.35.

## Results

A total of 49,341 adolescents aged 14–17 years died by suicide in India between 2014 and 2019. Of these, 23,075 were male and 26,266 were female. Over the study period (2014–2019), the national suicide rate among adolescents aged 14–17 years in India ranged between 7.12 and 8.74 per 100,000 population (Table 1 and Figure 1). Suicide rates were consistently higher among females (ranging between 8.10 and 9.04 per 100,000 population) compared to males (ranging between 6.24 and 8.47 per 100,000 population) between 2014 and 2019 (Table 1 and Figure 1).

Between 2014 and 2016, adolescent suicide rates in India decreased from 8.74 to 7.12 per 100,000 population. This decline was observed in both females (decreasing from 9.04 to 8.10 per 100,000 population) and males (decreasing from 8.47 to 6.24 per 100,000 population) (Table 1 and Figure 1).

However, in the latter half of the study period (2017–2019), suicide rates increased slightly, rising from 7.51 to 7.82 per 100,000 population overall. During this period, female rates increased from 8.45 to 8.98 per 100,000 population, while male rates increased from 6.64 to 6.76 per 100,000 population (Table 1 and Figure 1). These national trends were also evident among less developed states, but not among more developed states (Table 1 and Figure 1).

Compared to the first half of the study period, significant increases in suicide rates were observed in the latter half among less developed states overall (RR = 1.06, 95% UI = 1.03–1.09) and among females in less developed states (RR = 1.09, 95% UI = 1.05–1.14) (Table 2).

**Table 1.** Suicide rates (crude) for India among adolescents (14–17 years) by sex, and less or more developed state groups: 2014–2019

| | 2014 | 2015 | 2016 | 2017 | 2018 | 2019 |
|---|---|---|---|---|---|---|
| | Rate (95% CI)[a] | Rate (95% CI) | Rate (95% CI) | Rate (95% CI) | Rate (95% CI) | Rate (95% CI) |
| **India** | | | | | | |
| Total (both sexes combined) | 8.74 (8.56–8.92) | 7.47 (7.31–7.64) | 7.12 (6.97–7.29) | 7.51 (7.34–7.67) | 7.62 (7.46–7.79) | 7.82 (7.65–7.99) |
| Female | 9.04 (8.78–9.31) | 8.42 (8.17–8.68) | 8.10 (7.85–8.35) | 8.45 (8.20–8.71) | 8.62 (8.37–8.88) | 8.98 (8.72–9.24) |
| Male | 8.47 (8.23–8.72) | 6.61 (6.40–6.82) | 6.24 (6.03–6.45) | 6.64 (6.43–6.86) | 6.71 (6.50–6.93) | 6.76 (6.54–6.98) |
| **Less developed states** | | | | | | |
| Total (both sexes combined) | 5.65 (5.46–5.85) | 4.96 (4.78–5.14) | 4.70 (4.53–4.88) | 4.97 (4.79–5.15) | 5.34 (5.16–5.53) | 5.99 (5.80–6.19) |
| Female | 5.66 (5.38–5.95) | 5.42 (5.15–5.70) | 5.22 (4.96–5.50) | 5.51 (5.24–5.78) | 5.91 (5.63–6.20) | 6.44 (6.15–6.73) |
| Male | 5.64 (5.38–5.91) | 4.55 (4.31–4.79) | 4.23 (4.00–4.46) | 4.48 (4.25–4.72) | 4.83 (4.59–5.07) | 5.59 (5.33–5.85) |
| **More developed states** | | | | | | |
| Total (both sexes combined) | 12.51 (12.19–12.83) | 10.57 (10.28–10.87) | 10.19 (9.90–10.48) | 10.74 (10.45–11.05) | 10.53 (10.23–10.83) | 10.21 (9.92–10.51) |
| Female | 13.09 (12.62–13.57) | 12.11 (11.66–12.57) | 11.68 (11.24–12.14) | 12.21 (11.76–12.68) | 12.10 (11.64–12.56) | 12.29 (11.83–12.77) |
| Male | 11.97 (11.54–12.41) | 9.16 (8.79–9.55) | 8.82 (8.45–9.20) | 9.39 (9.01–9.79) | 9.09 (8.71–9.48) | 8.30 (7.93–8.67) |

[a]Suicide rate (crude) per 100,000 population; 95% CI, 95% confidence interval.

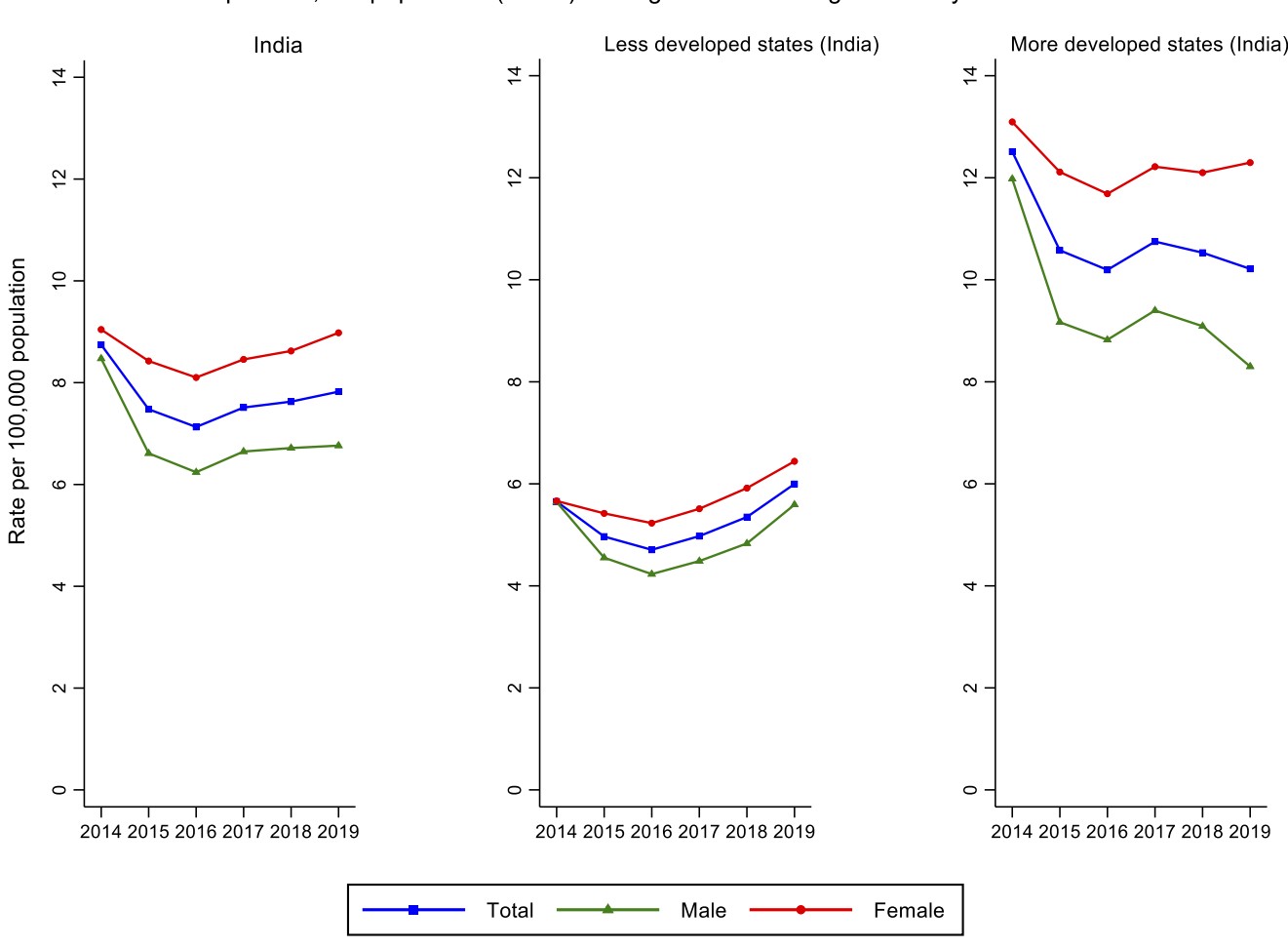

**Figure 1.** Suicide rates (crude) for India among adolescents (14–17 years) by sex, and less or more developed state groups: 2014–2019.

**Table 2.** Suicide rate ratios among adolescents (14–17 years) for 2017–2019 versus 2014–2016 with 95% UI by sex, and less or more developed state groups

| Total | RR 95% UI[a] (2017–2019÷2014–2016) |
|---|---|
| India | 0.98 (0.96–1.00) |
| Less developed states | 1.06 (1.03–1.09) |
| More developed states | 0.94 (0.92–0.96) |
| Male | |
| India | 0.94 (0.91–0.96) |
| Less developed states | 1.03 (0.99–1.07) |
| More developed states | 0.89 (0.86–0.92) |
| Female | |
| India | 1.01 (0.99–1.04) |
| Less developed states | 1.09 (1.05–1.14) |
| More developed states | 0.99 (0.96–1.02) |

[a]95% UI, 95% uncertainty interval.

Among females, states with the most pronounced increases in suicide rates during the latter half of the study period included Uttarakhand (RR = 2.83, 95% UI = 1.63–5.11), Punjab (RR = 1.84, 95% UI = 1.3–2.56), Jharkhand (RR = 1.73, 95% = UI 1.42–2.1) and Himachal Pradesh (RR = 1.55, 95% UI = 1.06–2.23) (Figure 2).

## Discussion

To the best of our knowledge, this is the first study to report on national- and state-level trends of suicide rates among adolescents (aged 14–17 years) in India based on the NCRB data. The study found that adolescent suicide rates in India averaged around 7.6 per 100,000 population over the study period. Notably, these suicide rates were consistently higher among females compared to males. Moreover, the study observed an increasing trend in suicide rates during the latter 3 years of the study period (i.e., 2017–2019) among less developed states, which was particularly driven by a rise in adolescent female suicide rates within those states.

Globally, adolescent suicide rates vary by sex, with conflicting evidence on whether males or females have higher rates. Some evidence highlights that female suicide rates exceed male rates among those aged 15–19 years (Naghavi, 2019), while other evidence suggests the opposite, especially in high-income countries (Glenn et al., 2020; Balachandran and Desai, 2024). These disparities likely stem from sociocultural factors and differing risk factors faced by males and females across and within countries. For example, in LMICs such as India, known risk factors for female adolescent suicides, such as mental disorders and social media influences (Bertuccio et al., 2024), might intersect with deeper structural and cultural inequalities. Additionally, variations in data sources and gendered patterns of underreporting contribute to systematic gaps in data on sex differences (Arya et al., 2021). This underscores the urgent need for improved data collection on adolescent suicides worldwide.

For India specifically, previous evidence indicates that suicide rates among females surpass those of males in the 15–19 age group (India State-Level Disease Burden Initiative Suicide Collaborators, 2018). Furthermore, while suicide rates among adolescent males in our study were somewhat comparable to average rates among males in high-income countries aged 15–19 years, female rates exceeded those rates by nearly threefold (Glenn et al., 2020). This trend is also reflected when comparing India's adolescent suicide rates with other countries in the Southeast Asia region. For example, male suicide rates in our study are similar to those in the Democratic People's Republic of Korea and Thailand but lower than in Sri Lanka and Nepal (WHO, 2024a). However, for females, the suicide rates in our study are two to six times higher than in all other countries in the region, except Sri Lanka, where the rate is slightly higher (WHO, 2024a).

Some risk factors for adolescent suicide, including in India, are substance abuse, academic pressures, mental disorders and family problems (Bilsen, 2018; Senapati et al., 2024). High suicide rates among young women in India might be associated with various factors, including gender-based discrimination, rigid patriarchal norms, early marriage and higher risk of depression (Petroni et al., 2015). However, there is a lack of understanding of how these factors specifically impact female adolescent suicides in India. For example, the NCRB data tabulates reasons for suicides under broad categories such as 'love affairs' and 'family problems', which fail to capture the complexity of underlying issues (NCRB, 2001–2022).

Adolescent suicide rates among males remained relatively stable over the study period. Globally, male adolescents are more likely to engage in high-lethality methods and have lower rates of help-seeking compared to females, which can contribute to higher suicide rates (Bilsen, 2018; Miranda-Mendizabal et al., 2019). In the Indian context, societal norms around masculinity may discourage boys from expressing vulnerability or accessing support (Dochania and Dochania, 2025). Risk factors, such as substance use and academic pressure, have been associated with male suicidal behavior in prior Indian studies (Senapati et al., 2024). However, the absence of an upward trend in male adolescent suicide rates may reflect either plateaued risks or limitations in how suicides among boys are reported or classified. Future research should explore the gendered dimensions of stigma, help-seeking and risk exposure to better understand the suicide trajectories of adolescent male suicides in India.

There was an increase in the suicide rates in the latter half of the study period (i.e., 2017–2019) in less developed states, especially among females. An increase in overall suicide rates in India post-2017 has been documented based on the NCRB data (Arya et al., 2022). While the exact reasons behind these increases remain unclear, it is plausible that improved reporting of suicides in certain states, particularly following the decriminalization of suicide in 2017 (Behere et al., 2015) might have contributed to these findings. Underreporting of suicides is known to be higher in less developed states compared to more developed ones, and higher among females than males (Arya et al., 2021). Therefore, any improvements in reporting, particularly in less developed areas, might disproportionately affect female suicide rates. The rise in adolescent female suicide in less developed states may also reflect growing tensions between increasing female autonomy and persistent restrictive gender norms in some parts of India (Singh et al., 2021; Anderson, 2024; Balachandran and Desai, 2024). Regardless of the reasons, the rise in suicide rates among adolescent girls in less developed states is concerning, especially given the resource constraints and limited access to mental health services in these regions compared to some of the more developed states (Nandy, 2019; Suhas et al., 2023).

Suicide data from more recent years will be helpful in ascertaining changes in suicide rates among adolescents in India (e.g., changes in adolescent suicide rates during the coronavirus disease 2019 [COVID-19] pandemic years). However, investigating

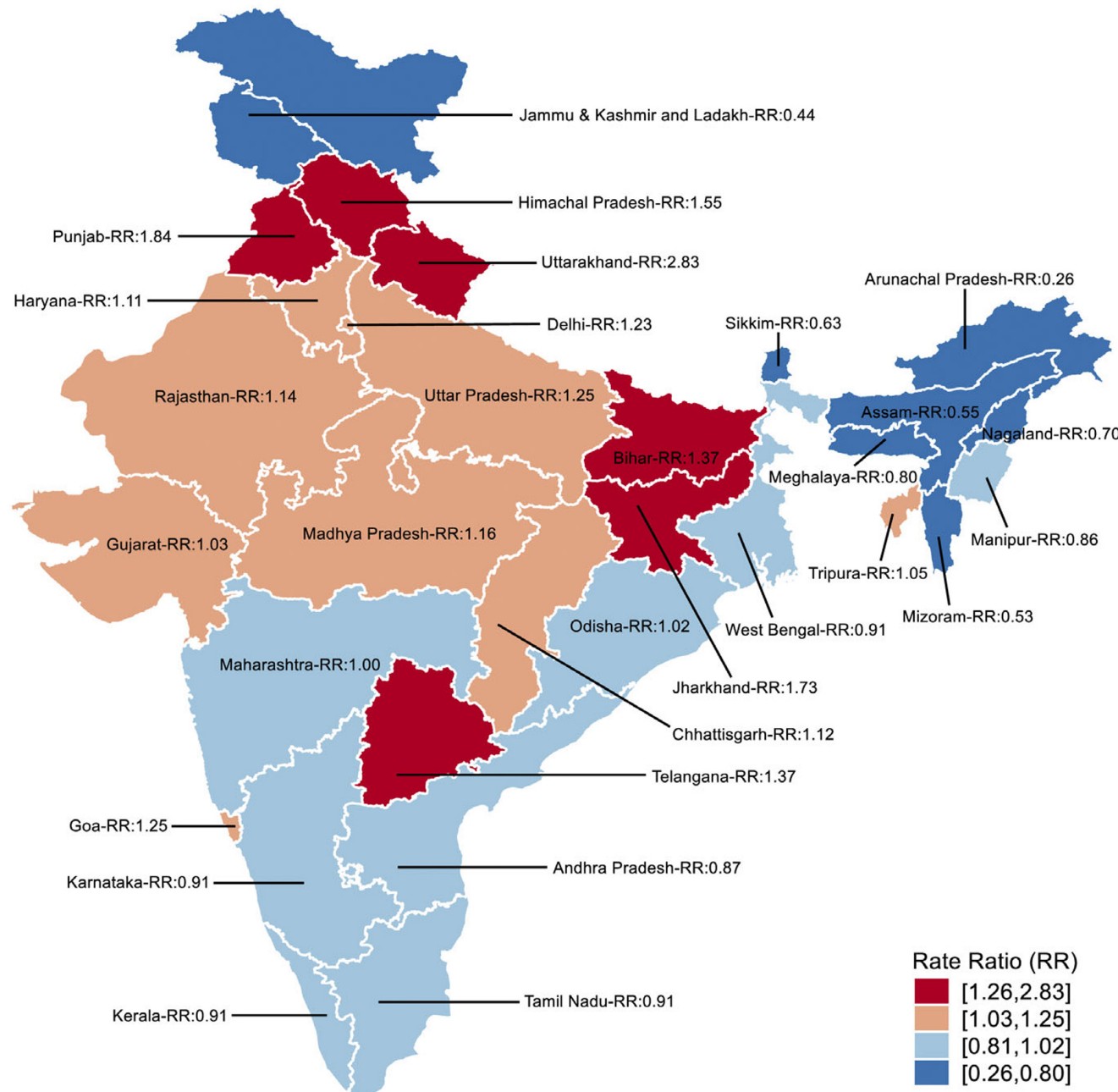

**Figure 2.** Suicide rate ratios for 2017–2019 versus 2014–2016: female.

adolescent suicide rates in India post-2019 is currently challenging due to limitations in the NCRB data. Recent NCRB reports provide suicide statistics by age group at the national level, lacking detailed state-level breakdowns. Additionally, the introduction of the age category '18 and below' (replacing 'up to 14 years' and '14–18 years' age categories) complicates the estimation of suicide rates among adolescents (NCRB, 2001–2022). To improve the utility of data, the NCRB should consider reverting to publishing suicide statistics by state, sex and age group, specifically breaking down adolescent age groups into 10–14 and 15–19 years. A breakdown of this data by suicide method would also be beneficial. For example, a multi-country study (including India) found the greatest proportional reductions in suicide among young people, especially young females, following bans on specific pesticides (Schölin et al., 2023).

Additionally, the release of de-identified individual-level data on variables, such as educational level, marital status and economic status, would help determine risk factors for adolescent suicides. Information on potential contributing factors, such as school-related stressors (e.g., academic pressure and bullying), and individual-level psychological characteristics, such as impulsivity, which are known to play a significant role in adolescent suicidal behavior (Auerbach et al., 2017; Steare et al., 2023), would also be helpful. Collectively, these improvements in data stratification would enable more precise analysis and better-informed interventions.

The findings of this study can inform policymaking, including through the IAHS and the NSPS. It has been suggested that the IAHS is best suited to address adolescent mental health issues, including suicidal behavior (Roy et al., 2019). Within the IAHS

framework, the theme 'enhance mental health' highlights the importance of addressing adolescent mental health through counseling services and the introduction of life skills education, an intervention endorsed by the WHO for suicide prevention among adolescents (Ministry of Health and Family Welfare, 2014b; WHO, 2021). The NSPS identifies developing community resilience and reducing stigma around suicidal behaviors as a priority, with suggested activities, including gatekeeper training for teachers, integrating mental health and substance use prevention into school curricula and supporting adolescents' psychological development (Ministry of Health and Family Welfare, 2022).

However, the IAHS and the NSPS can further help address adolescent suicides in India. For example, the difference in adolescent suicide rates among males and females highlights the gendered nature of suicidal behavior and its determinants. However, as highlighted by a recent review, the IAHS does not explicitly identify pathways or interventions that can challenge gender norms to tackle gender inequities in early adolescence (Dandona et al., 2024). Pilot programs conducted in Indian schools, such as the Gender Equity Movement in Schools (GEMS), which aim to sensitize adolescent boys to the issues of gender-inequity and violence, could be implemented at a wider scale (Achyut et al., 2011). School-based interventions are especially important as they, along with digital interventions, are the platforms known to attract the highest adolescent engagement (Roy et al., 2019). However, this does not imply ignoring the more disadvantaged adolescents who are neither in school nor active on digital platforms. For example, a brief mental health and resilience intervention targeting young female school leavers in an urban slum in North India reported improvements in depression, anxiety and attitudes toward gender equality among the participants (Mathias et al., 2018).

The IAHS neither mentions suicide surveillance nor outcome indicators related to suicidal behavior, making it difficult to assess the extent of adolescent suicide or gauge the impact of policies addressing this issue. The NSPS can play a vital part in addressing these gaps. The NSPS highlights suicide surveillance as one of the priority areas. Moving forward, the NSPS should prioritize the surveillance of adolescent suicidal behavior. The NCRB data can be the primary source for this information, while monitoring of self-harm presentations in hospital settings can also be a useful source of surveillance. The IAHS can also play a role in surveillance by adopting the Global Action for Measurement of Adolescent Health (GAMA) indicators for measuring mental health among adolescents in India. Developed by the WHO and the United Nations, GAMA provides a comprehensive set of indicators to enhance the measurement and understanding of adolescent health globally, including mental health (WHO, 2024b). Incorporating these indicators in school-based surveys, for example, can help collect standardized data on mental health, including suicidal behavior, across states within India (Rizvi et al., 2024). Ongoing assessment of data collected through NCRB, hospital presentations and school-based surveys can act as outcome indicators related to suicidal behavior. Thus, the alignment of the IAHS and the NSPS goals can help address adolescent suicidal behavior in India.

There are some limitations to this study. First, although the NCRB data provide comprehensive national coverage and are the primary source of official suicide statistics in India, they likely underreport suicide counts due to stigma, legal issues and social pressures (Arya et al., 2021). The NCRB data are not designed for active surveillance and rely on passive reporting through First Information Reports filed by family members at local police stations, which further affects data completeness. Second, due to data

constraints, this study could not provide suicide rates for the traditional adolescent age groups: 10–14 and 15–19 years, which would have allowed for better comparison with other countries and data sources. Third, the nonavailability of recent data meant that the study could not report on suicide rates for more recent years, including those during the COVID-19 pandemic. Despite these limitations, this is the first study to report suicide rates among adolescents aged 14–17 years in India based on the NCRB data, both at the national and state levels, broken down by sex.

In conclusion, suicide rates among adolescents aged 14–17 years in India are notably higher among females compared to males, with increasing rates particularly in less developed states. The IAHS and the NSPS will play vital roles in addressing adolescent suicidal behavior in India, including comprehensive surveillance and implementation of targeted and inclusive prevention strategies, especially those addressing the gendered aspects of adolescent suicide. Additionally, the NCRB should enhance data reporting, disaggregating adolescent suicide data by state, sex and age groups of 10–14 and 15–19 years. These steps are crucial for effectively mitigating this significant public health issue.

**Open peer review.** To view the open peer review materials for this article, please visit http://doi.org/10.1017/gmh.2025.10044.

**Supplementary material.** The supplementary material for this article can be found at http://doi.org/10.1017/gmh.2025.10044.

**Data availability statement.** Suicide count data are available upon request from the authors.

**Acknowledgments.** The authors would like to sincerely thank the National Crime Records Bureau (NCRB) for providing them with unpublished suicide data between 2016 and 2019.

**Author contribution.** V.A. led the data curation, conceptualization, investigation, formal analysis, visualization, data interpretation, writing and revising. C.T. supported data analysis and revision. G.A.K. provided GBD population numbers for Indian adolescents and supported revising. R.D., J.P. and G.A. supervised, supported conceptualization, visualization, data analysis and revision. S.O., P.B., M.S., A.P. and L.V. supported revision.

**Financial support.** The authors declare that no specific funding has been received for this article.

**Competing interests.** The authors declare none.

**Ethics statement.** No institutional ethics approval was required since the study was based on de-identified, secondary-level data.

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
