## [Reviewer Report]

Page 4 line 1 – I thought motor vehicle collisions were the leading cause of death among adolescents. Check this.

Page 6, lines 12-15: Why not conventional confidence intervals? You need some justification.

I found the long discussion section digressive. There ws much too much in this section to qualify as a discussion of the empirical findings and much too little to be a serious discussion of the many different intervention possibilities considered. My inclination would be to delete much of this. It comes across as superficial and in some places naïve.

---

## [Reviewer Report]

Thanks to the author for his work and contribution, there are a few comments on the manuscript:

1.Introduction

1.1 In the Introduction, the authors emphasize the key significance of adolescent suicide, but fail to describe the harm and social impression of adolescent suicide in India, and the research significance and background are not enough. Can the authors supplement the relevant content?

1.2 In addition, can the author supplement the theoretical basis of the study?

1.3 The author mentions two national strategies for suicide prevention, but the effectiveness of these strategies should be briefly explained to highlight the importance of this study.

2. Methods

2.1 The “Method” section should briefly describe the design and framework of this study

2.2 What is the scientific basis for the authors comparison of “rate ratios (RRs) of crude suicide rates” by mentioning “by dividing crude suicide rates of the last three years of the study period (i.e., 2017-2019) by the first three years of the study period (i.e., 2014-2016)”? Is it statistically significant?

2.3 The author should report the relevant content of sample size in the “method” section, such as the calculation method of sample size and the required sample size of the study, so as to prove that the sample size of this study and the statistical analysis results based on the sample size are representative.

2.4 The author divides the research period into two periods: 2014-2016 and 2017-2019. There is insufficient scientific basis for such division. What is the purpose and significance of such division?This division could potentially account for any

2.5 The authors only calculated the crude mortality rate and its 95% confidence interval, the crude mortality ratio and its 95% uncertainty intervals (UIs), and stratified analysis was conducted only by sex and region (Less developed states and More developed states). The statistical method was relatively simple, and it is difficult to have convincing results from this analysis.

3. Results

3.1 The author should show the total number of samples obtained in the “Results” section, as well as the basic demographic and sociological characteristics of the total number of samples obtained.

3.2 The authors describe the trend of adolescent suicide rates from 2014 to 2019 and the rate stratified by sex, but only descriptive statistics are performed, with a simple list of changes in rates without in-depth statistical inference. It is not possible to confirm whether these changes and differences are statistically significant.

4. Discussion

4.1 Due to the above defects in the research methods and the inadequacy of the results report, the scientific significance of the discussion section is insufficient.

4.2 The authors discussed the factors related to suicide in the “Discussion” section, but did not address this aspect of the study.

5.Overall, this manuscript, while presenting trends and changes in adolescent suicide rates in India from 2014-2019, is not significant in terms of academic value and is slightly less innovative and scientific.

---

## [Reviewer Report]

Key Points

• The study contributes valuable knowledge to the limited literature on suicide in low- and middle-income countries (LMICs), especially India.

• The work is highly significant and fills a critical gap, particularly in the emerging research on adolescent suicide in Asia. It provides important insights into a vulnerable population using novel/emerging methods.

Major Issues and General Comments

• The term “suicide” is used broadly throughout the abstract and manuscript. Suicide encompasses various aspects, including suicidal ideation, planning, attempts, self-harm, and death by suicide. It is crucial to specify which aspect is being addressed or use more precise terminology (e.g., suicidal deaths).

Minor Issues

Introduction

• It would be helpful to discuss suicide rates globally, in Asia, and in Southeast Asia more broadly before focusing specifically on rates in India.

• The authors mention, “India, which accounts for approximately 28% of global suicides (India State-Level Disease Burden Initiative Suicide Collaborators, 2018),” but I am a little skeptical about this given that there was an article that ranked India 43rd in the world (https://pmc.ncbi.nlm.nih.gov/articles/PMC3554961/). Maybe double-check across various sources that this is accurate? I couldn’t find this exact number in the citation presented.

• The sentence, “The two policies best suited to address adolescent suicides in India are the National Suicide Prevention Strategy (NSPS) (Ministry of Health & Family Welfare, 2022) and the India Adolescent Health Strategy (IAHS) (Ministry of Health & Family Welfare, 2014a). The NSPS mentions that most suicides in India are and trends,” is a bit confusing. Are these policies or datasets? They are discussed as though they are datasets but the authors say that they are “policies” in the manuscript.

• It would strengthen the rationale to include an overview of prior suicide research in India—particularly among adolescents—and clearly describe where the gaps lie. I noticed several reviews already exist in India on adolescent suicidal behaviors and risk factors, so it would be useful to specify how your study builds on and extends this work. As written, the rationale for the study is not yet well established.

• It would also be helpful to fully explain why examining “trends in adolescent suicide rates in India from 2014 to 2019 at both national and state levels, disaggregated by sex” is important.

• Consider defining suicide in the introduction and contextualizing the term for your specific study.

• Recommend using “less/more resourced” instead of “less developed” and “more developed” to improve precision and sensitivity.

Methods

• In the methods section, the authors begin by discussing the National Crimes Records Bureau (NCRB) dataset, but then abruptly transition to the Global Burden of Disease (GBD) in lines 6–34. The connection between these two datasets and their respective roles in the analysis is unclear and would benefit from clarification.

• Please add a sentence explicitly stating what years are being analyzed for each dataset. While the types of data are described, it is difficult to parse the exact time periods and the connection between datasets.

• Is the data nationally representative?

• If there were any issues with missing data, please discuss how these were addressed.

Results

• Consider clarifying the following sentence, especially regarding the range presented: “Over the study period, suicide rates among adolescents aged 14–17 years in India ranged between 7.12 and 8.74 per 100,000 population (Table 1; Figure 1).” Are these ranges across states, or over time? Clarification would aid comprehension and ensure consistency with how results are presented in the abstract as well.

Discussion

• The sentence, “These disparities likely stem from socio-cultural factors and differing risk factors faced by males and females across and within countries,” introduces a major theme of the paper. It would be helpful to expand on these risk factors, particularly in the context of India. Where possible, compare findings to global literature and briefly explore why there may be an increase in suicide among adolescent girls in lower SES states. While you touch on this in lines 36–53, expanding further would enrich the discussion.

• Consider discussing other limitations, such as the lack of focus on school-related factors, which may be particularly relevant for this age group. Impulsivity is another factor worth mentioning, given its importance in adolescent suicidal behavior.

• Otherwise, great job and interested to see the paper move forward.

---

## [Reviewer Report]

Overall statement:

I have found the manuscript is well-organized, and the objectives, method, and results sections are aligned with each other. Based on my review, I would recommend it for publication with minor revisions. I hope, the authors will address the following suggestions to enhance the quality of this manuscript.

Method: You have segregated the states as less developed and more developed. Could you please include an explanation of how this distinction is aligned with the objectives of this inquiry.

Though you have already declared that you have analyzed secondary sources data, it is essential for the researchers to ensure that ethical standards were upheld during the original data collection process. Hence, I request you to explain the following concerns:

1. How did the NCRB collect data? Did their data represent a primary source?

2. How did they obtain permission from participants to publish the data?

3. How did they maintain ethical standards during the collection of data, as suicide and suicidal attempts are highly stigmatized behaviors in many cultures and countries like India?

The last concern regarding the data source: NCRB collects suicide data, and NCRB is the National Crimes Records Bureau. Is suicide considered a crime in India? If not, why does the National Crime Board collect this stigmatized and sensitive data? What is the role of NSPS and IAHS in this context?

Discussion: Your discussion section has articulated the findings very well. I have noticed that you have intensively interpreted the findings on female adolescents. The findings have significant implications for policies and the practical field; hence, I would suggest including one or two paragraphs which explain the suicidal risk factors for male adolescents, and what can the be the potential causes for stable rates of male suicide over the years.

Limitation: Data from secondary sources have been analyzed; hence, how reliability and validity of the findings can be ensured needs to be included in the methodology section; otherwise, I would suggest including it in the limitation section.

Thank you.

---

## [Reviewer Report]

Thank you for the opportunity to review this important and well-executed paper. I have no further comments at this stage and commend the authors for thoroughly addressing the earlier feedback. I recommend the manuscript be accepted for publication.

It would be valuable to engage the IAHS and the NSPS after publication to further align efforts and advance the implementation of gender-responsive and surveillance-driven interventions. This paper provides a strong foundation for those next steps.